# A large scale hearing loss screen reveals an extensive unexplored genetic landscape for auditory dysfunction

Michael R. Bowl

The developmental and physiological complexity of the auditory system is likely reflected in the underlying set of genes involved in auditory function. In humans, over 150 non-syndromic loci have been identified, and there are more than 400 human genetic syndromes with a hearing loss component. Over 100 non-syndromic hearing loss genes have been identified in mouse and human, but we remain ignorant of the full extent of the genetic landscape involved in auditory dysfunction. As part of the International Mouse Phenotyping Consortium, we undertook a hearing loss screen in a cohort of 3006 mouse knockout strains. In total, we identify 67 candidate hearing loss genes. We detect known hearing loss genes, but the vast majority, 52, of the candidate genes were novel. Our analysis reveals a large and unexplored genetic landscape involved with auditory function.

#A full list of authors and their affliations appears at the end of the paper

Hearing impairment is the most common sensory deficit in the human population. According to figures from the World Health Organisation, there are currently around 360 million people worldwide living with mild to profound hearing loss. Approximately half of these cases have a genetic basis, with the hearing loss either occurring as an isolated condition (non-syndromic, 70%) or presenting with additional phenotypes (syndromic, 30%). Around 150 non-syndromic loci have been identified in humans (hereditaryhearingloss.org) and over 400 genetic syndromes are known that include hearing loss[1, 2]. To date, around two-thirds of the genes for non-syndromic hearing loss loci are known[3, 4], and, while the causative genes for several of the well-known hearing loss syndromes have been identified[5], the vast majority of genes underlying syndromes with hearing loss are unknown. As such, we are far from having a complete understanding of the genetics underlying hearing function. Importantly, we do not know the extent of the mammalian gene set that impacts auditory function. Mouse genetics has played an important role in our understanding of the development and functioning of the mammalian auditory system[6, 7]. Utilising forward-genetic screens, many hearing loss mouse mutants have been identified[8, 9], and characterisation of these has enabled genes critical for hearing function to be elucidated. Allied with gene-driven approaches, mouse mutants continue to help identify the molecular mechanisms and physiological bases of human hearing impairment. Moreover, the availability of a disease model that allows for the study of pathological changes occurring within the cochlea is invaluable for understanding disease onset and progression.

The International Mouse Phenotyping Consortium (IMPC) aims to generate and phenotype a null mutant for every gene in the mouse genome (www.mousephenotype.org)[10, 11]. Until recently, IMPC has generated mouse mutants using principally the International Knockout Mouse Consortium (IKMC) tm1b null allele (see Methods section and Skarnes et al.[12]). Homozygous viable mutants, or heterozygotes in the case of homozygous lethal lines, enter an extensive adult phenotyping pipeline that includes a range of phenotyping tests generating broad-based phenotype information across diverse physiological systems, including hearing[10]. To date, the IMPC has generated over 5800 genotype confirmed mutant strains and of these, 3006 have been subject to Auditory Brainstem Response (ABR) testing to determine their hearing thresholds.

Here, using a robust statistical approach coupled with manual curation, we analyse the ABR data set generated from the 3006 mutant lines and identify 67 that show elevated hearing thresholds compared to wild-type control mice, matched for strain, gender and age. Our findings identify a significant number of candidate hearing loss genes that have not previously been reported, which not only increases our knowledge of the auditory genetic landscape and the molecular mechanisms required for hearing, but also provides a fund of new genes for the many unidentified loci in the human population.

## Results

**Auditory phenotyping**. In order to identify genes required for hearing function, the consortium uses an ABR test in the adult pipeline at week 14 that assesses hearing at five frequencies—6, 12, 18, 24 and 30 kHz. The consortium aimed to analyse a minimum of four mutant mice for each gene and, in most cases, both mutant males and females were analysed. Data from each IMPC centre is uploaded to the IMPC Data Coordination Centre, where it is subject to data validation and quality control (QC) procedures prior to statistical analysis and public access (www.mousephenotype.org).

**Statistical analysis**. Statistical analysis for a high-throughput project such as IMPC requires a robust generic analysis pipeline to produce statistically significant and biologically relevant results. IMPC uses the 'PhenStat' package[13], which employs a number of different statistical techniques appropriate to the phenotype data being examined.

To date for ABR data, a mixed model approach has been employed in IMPC, using sex and genotype as fixed parameters and date-of-procedure as a random term (in order to allow for any batch effects). The genotype parameter is tested for its significant effect on the model, and this p-value is returned for the genotype effect within the data. However, mixed models cannot be fitted when the target values are all the same, which occasionally happens in ABR data (e.g., when every mutant animal is profoundly deaf at all frequencies). Thus we adopted a different method to undertake a more robust analysis of the ABR data set.

The PhenStat package includes a 'reference range plus' function[13]. Within this function, a reference range is created by ordering all of the baseline data points and creating a 98% reference range. This is achieved by taking the 1% percentile and 99% percentile data values, between which 98% of all values lie. The reference range is a non-parametric technique that makes no assumptions about the data distribution. The number of observations within and without this range is counted for the baseline and the mutant data values. A two-way contingency table is thus created. A Fisher's exact test is then used to compare the baseline values against the mutant value, producing a p-value that reflects whether the two sets of counts are from the same data set. For our analysis we employed all the available IMPC ABR mutant data, and C57BL/6N wild-type data. For each genotype, the mutant and matched wild-type data were used to create reference ranges, build contingency tables, and calculate p-values.

Using this approach, the available ABR data from 3006 lines yielded 328 candidate hearing loss genes (Supplementary Data 1). The genotype p-values provide an initial indicator of possible biological significance. As such, the ABR phenotyping data for each of these 328 lines underwent expert-led manual curation to assess phenodeviance and identify a robust data set of candidate hearing loss genes with a low false discovery rate.

**Manual curation**. The ABR threshold data for each of the 328 individual mutant strains were examined to ascertain the spread of data points in relation to their respective centre wild-type mean. Strains showing discordant thresholds within sex were removed from the list. In addition, strains that exhibited elevated thresholds at only one middle-range frequency (12, 18 or 24 kHz) were also removed from the list. Employing these criteria led to 261 strains being removed from further analyses. Strains that showed highly concordant elevated thresholds at either, low frequency (6 kHz), high frequency (30 kHz) or across two or more frequencies, were deemed to be true hearing-impaired mutants (65 strains). In addition, strains with hearing thresholds that were discordant between sex, but concordant within sex were also deemed to be true hearing-impaired mutants (2 strains). Using these criteria, our analyses generated a list of 67 mutant strains that we deemed to be robust auditory pheno-deviants (Supplementary Table 1; Fig. 1). The majority of these 67 candidate hearing loss genes were identified from homozygous viable mutants (Supplementary Table 1). However, a small number of heterozygous mutants (eight in total) from homozygous lethal lines showed hearing impairment.

The stringent critical p-value and manual curation parameters were selected to allow production of a robust gene list with few false positives. However, we recognise that our analyses will likely

generate some false negatives. To test this, we examined the set of 261 excluded genes across all relevant gene function databases for evidence of an association with hearing loss. PubMed, OMIM and MGI searches demonstrated that only nine had evidence for an auditory phenotype in either human or mouse (Supplementary Table 2). Six of these genes, however, showed association with hearing loss in human patients only and studies of mouse mutants have failed to uncover an auditory phenotype. This may reflect a difference in the role of these genes between mouse and human, or alternatively, the impact of genetic background on penetrance.

**Candidate hearing loss gene classification.** Dependent on their auditory profile, we assigned each of the 67 candidate genes to four broad classes of hearing loss using the following criteria: only the higher frequencies affected—high-frequency hearing loss (13 genes); only the lower frequencies affected—Low-frequency

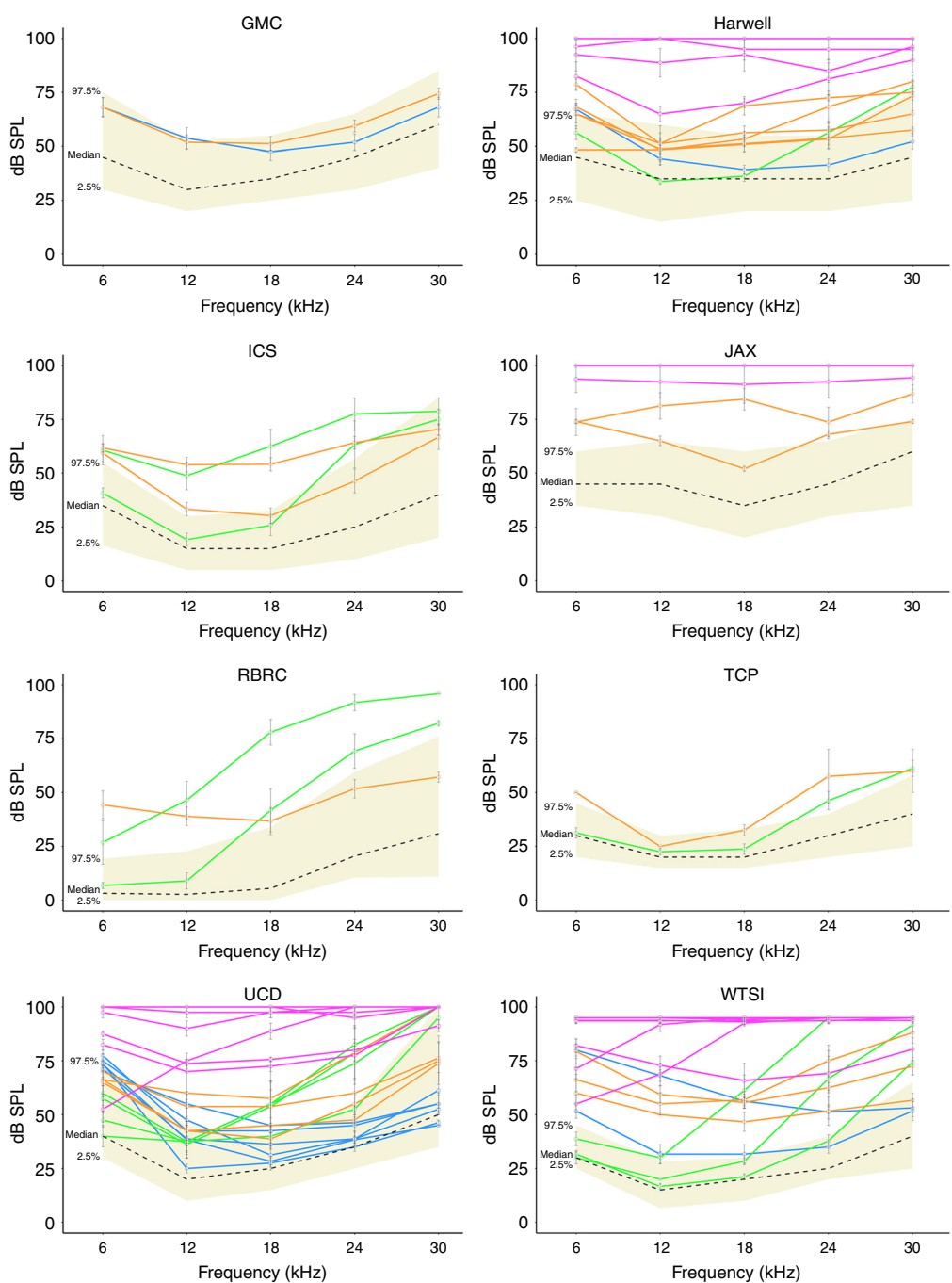

**Fig. 1** Summary audiograms for hearing loss genes identified by individual IMPC centres. At each centre auditory thresholds (dB SPL) were assessed at five frequencies—6, 12, 18, 24 and 30 kHz. Each graph, displaying the data from a single IMPC centre, portrays a reference range (*yellow shaded area*) and median (*dashed line*) for control wild-type animals. The colour of each audiogram line relates to the classification of hearing loss as shown in Fig. 2 and listed in Table 1. Severe, *magenta*; mild, *orange*; high frequency, *green*; low frequency, *blue*. Phenotyping centres: GMC, Helmholtz Zentrum Munchen; Harwell, MRC Harwell; ICS, Institut Clinique de la Souris; JAX, Jackson Laboratories; RBRC, RIKEN Tsukuba Institute, BioResource Center; TCP, The Centre for Phenogenomics; UCD, University of California, Davis; WTSI, Wellcome Trust Sanger Institute

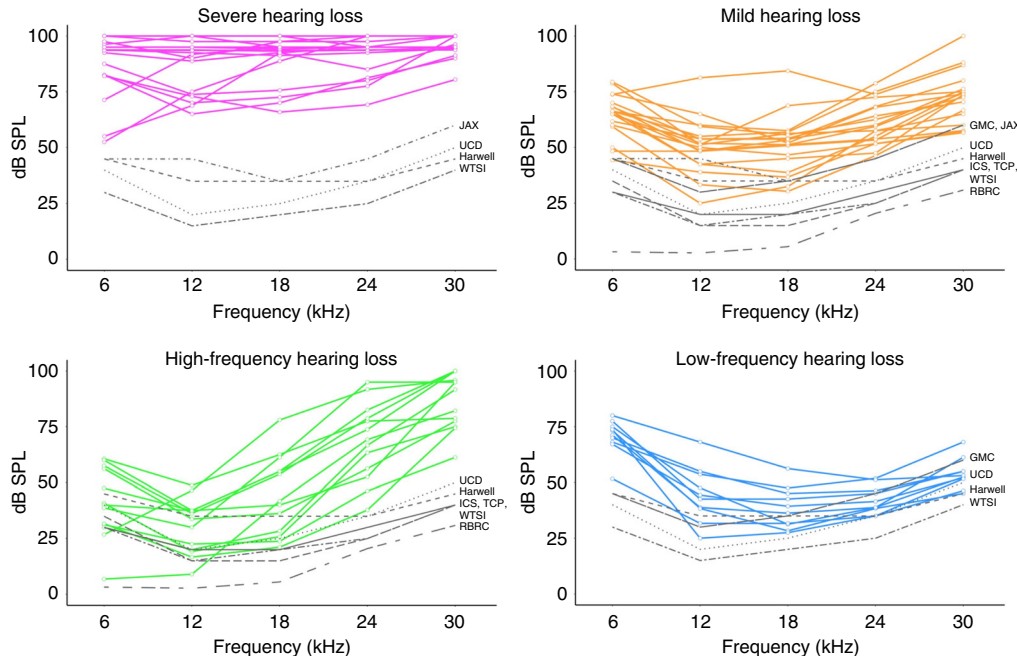

**Fig. 2** Summary audiograms for the 67 hearing loss genes identified within the IMPC. ABR screen assessing auditory thresholds (dB SPL) at five frequencies—6, 12, 18, 24 and 30 kHz. The genes are divided into four broad categories of hearing loss—severe (*magenta lines*), mild (*orange lines*), high frequency (*green lines*) and low frequency (*blue lines*). In each category, the median of auditory thresholds for control mice for each of the contributing centres to genes in that category is shown (*black lines*). GMC, Helmholtz Zentrum Munchen; Harwell, MRC Harwell; ICS, Institut Clinique de la Souris; JAX, Jackson Laboratories; RBRC, RIKEN Tsukuba Institute, BioResource Center; TCP, The Centre for Phenogenomics; UCD, University of California, Davis; WTSI, Wellcome Trust Sanger Institute

hearing loss (10 genes); if three or more frequency thresholds are ≥45 dB above the centre median—severe hearing loss (25 genes); and, if <3 frequency thresholds are ≥45 dB above the centre median—mild hearing loss (19 genes) (Fig. 2; Table 1; Methods section). PubMed, OMIM and MGI searches indicated that 15 of the 67 genes were known hearing loss-associated genes. However, despite these database searches 52 genes had not been associated with an auditory phenotype in either mouse or human. These genes were found in all four classes of hearing loss (Fig. 2; Table 1).

Our investigation of false negatives (see above) allows us to calculate a false omission rate for loci that may have been missed by our stringent manual curation of the set of 261 genes for which there was initial evidence of an auditory phenotype. Overall, the false omission rate for known loci is 3/261 or around 1%. Extrapolating to novel loci, based on the ratio of novel to known loci (52/15) within the set of 67 candidate hearing loss genes, we surmise that around 10 (i.e., 3 × 52/15) additional loci would be undiscovered from the set of 261 genes. This provides an estimate of 13/261 for the false omission rate.

The screen we have undertaken is effectively unbiased for both the developmental and physiological origin of the auditory phenotype, as well as for syndromic features that might be associated with the observed hearing loss. Without further investigation of individual mutants, we are unable to distinguish between hearing loss due to a deficit at the level of the central auditory pathway, a sensorineural origin in the inner ear, or due to conductive hearing loss due to middle ear problems, such as otitis media. However, we can take advantage of the comprehensive phenotype assessment that each mutant undergoes in the IMPC phenotyping pipeline, and are able to assess the prevalence of additional phenotypes in the hearing loss mutants. In Supplementary Table 3, we present a summary of the additional phenotypes observed for the 67 mutants, both the known and

novel candidate hearing loss genes. Sixty of these mutants (~90%) show multiple phenotypes (ranging from 1 to 64 additional significant phenodeviant parameters). We observed that a large number of mutants were positive for the Combined SHIRPA and Dysmorphology (CSD) test (33 out of 67, ~50%). Our analysis indicates that this is due to the absence of a Preyer reflex in response to a click box test (see Methods section), which is a strong indicator of a profound auditory deficit. Indeed, the majority of mutants classified as severe hearing loss were reported to have an absent Preyer reflex (20 out of 25, 80%) (Supplementary Table 3). Auditory mutants can sometimes also display a vestibular component to their abnormal phenotype. The CSD test includes a visual inspection for behavioural signs that may indicate vestibular dysfunction, e.g., head bobbing or circling. Only 3 of the 67 candidate hearing loss mutants were identified as having a head bobbing/circling phenotype: *Elmod1*, *Myo7a* and *Ush1c*, which are known hearing loss genes. The respective allelic mutants *roundabout*, *shaker-1* and *deaf circler* have previously been reported to exhibit vestibular dysfunction[14–16]. However, aside from the CSD test, there is no obvious pattern to the prevalence of additional phenotypes observed in the mutants tested. Of the 60 mutants that display additional phenotypes: 44 have 1-to-10 additional phenodeviant parameters; 10 have 11-to-20 additional phenodeviant parameters; 4 have 21-to-30 additional phenodeviant parameters; and 2 have >50 additional phenodeviant parameters detected. In contrast, seven of the mutants do not show any significant phenodeviance for any other phenotype parameter tested.

**Protein network interactions and pathways.** The products of the 52 novel candidate hearing loss genes encompass a very wide range of functions, from structural proteins to transcription factors, which reflects the complexity and cell type diversity of the auditory system. We assessed for potential interactions between

**Table 1 Summary of known and novel hearing loss genes identified in the IMPC hearing loss screen**

| Type of hearing loss | Known genes | Novel genes |
|---|---|---|
| Severe (all frequencies) | Adgrv1 | A730017C20Rik |
| | Cib2 | Duoxa2 |
| | Clrn1 | Eps8l1[a] |
| | Col9a2 | Klc2 |
| | Elmod1 | Nedd4l |
| | Gipc3 | Nptn[b] |
| | Ildr1 | Slc5a5 |
| | Marveld2 | Spns2[b] |
| | Myo7a | Tmem30b |
| | Ocm | Tmtc4 |
| | Otoa | Tox |
| | Tprn | Zfp719 |
| | Ush1c | |
| Mild (all frequencies) | Srrm4 | Acvr2a |
| | | Adgrb1 |
| | | Ankrd11 |
| | | Ap3m2 |
| | | Ap3s1 |
| | | Baiap2l2 |
| | | Ccdc92 |
| | | Cyb5r2 |
| | | Gga1 |
| | | Mpdz |
| | | Myh1 |
| | | Nisch |
| | | Odf3l2 |
| | | Slc4a10[b] |
| | | Tram2 |
| | | Ube2b |
| | | Ube2g1 |
| | | Vti1a |
| High frequency | | Aak1 |
| | | Acsl4 |
| | | Ahsg |
| | | Ccdc88c |
| | | Dnase1 |
| | | Emb |
| | | Ewsr1 |
| | | Gpr152 |
| | | Ikzf5 |
| | | Nin |
| | | Phf6 |
| | | Ppm1a |
| | | Wdtc1[a] |
| Low frequency | Gata2 | Atp2b1 |
| | | B020004J07Rik |
| | | Gpr50 |
| | | Il1r2 |
| | | Klhl18 |
| | | Med28 |
| | | Nfatc3 |
| | | Sema3f |
| | | Zcchc14 |

Genes are categorised into four broad categories of hearing loss
[a]Genes where only females are affected
[b]Genes that have been recently published

functions revealed. We also undertook a gene ontology analysis of both the novel and known IMPC hearing loss gene sets (Supplementary Table 4). While there was no significant gene ontology enrichment in the 52 candidate hearing loss genes, we found a wide range of predicted and enriched functions when the novel and known hearing loss genes were amalgamated and analysed together. We observed that in many cases while the candidate hearing loss genes were represented at the highest ontological level, this declined markedly at associated lower levels (Fig. 4), again reflecting the novel and unexplored functionality of this set of genes and potential new insights into gene function within the auditory system.

For three genes, recent reports from the Consortium confirm the hearing loss in mutant mice—Spns2[17], Nptn[18, 19], Slc4a10[20]—that emerged subsequent to identification in the IMPC screen. For example, the identification via the IMPC catalogue of Neuroplastin (Nptn) as a novel hearing loss gene has recently been extended in a report that demonstrates a role for this protein in synaptogenesis in inner hair cells in the organ of Corti in the mouse[18]. Neuroplastin, along with Embigin (Emb) and Basigin (Bsg), form a small family of neural cell adhesion molecules, belonging to the Ig superfamily. It is noteworthy that we report here Emb as a candidate hearing loss gene demonstrating high-frequency hearing loss, indicating that this family of neural CAMs merits further investigation as to their wider role in the auditory system.

## Discussion

We have assessed 3006 mouse mutants for hearing using an ABR test, and identified a large number of candidate hearing loss genes. After automated statistical analysis of the data set followed by manual curation, we uncovered a set of 67 mutants with hearing impairment. Of the 67 genes, 15 were known hearing loss loci, while the vast majority, 52, were novel candidate hearing loss genes that had not previously been associated with hearing loss.

Each of the 52 candidate hearing loss genes will merit further investigation. Some of the candidate genes are known to be expressed in the inner ear (e.g., Atp2b1) or to play a role in some aspect of hair cell function in the cochlea (e.g., Sema3f and sensory hair cell innervation)[21, 22]. However, for the majority of candidate loci, it will be important to further explore expression and function in the auditory apparatus. In particular, an ABR screen is relatively unbiased and will reveal auditory phenotypes that arise due to conductive hearing impairment as well as sensorineural hearing loss, or possibly a combination of both. We cannot rule out that some of the candidate hearing loss genes impact on middle ear function, or cause otitis media. This class of hearing loss genes is unlikely to be represented by mutations with severe hearing loss or high-frequency hearing loss, and would be more likely to be present in the class of genes with low-frequency or mild hearing loss. The mouse has proved an excellent model for the identification of genes involved in chronic otitis media[23–25] and it will be interesting to determine if the set of candidate hearing loss genes uncovered here further extends our knowledge of the genetic pathways involved in this form of hearing loss.

The unbiased nature of the IMPC phenotyping pipeline encompassing a wide range of phenotyping tests covering various biological systems allowed us to explore the pleiotropic state of the mutants that we have discovered that may inform us about the syndromic nature of the candidate hearing loss genes we have identified. In the human population, many forms of hearing loss present as a component of a wider genetic syndrome with a range of phenotypes. It is quite likely that some non-syndromic forms of hearing loss may also have additional unrecognised phenotypes. Large-scale studies of mutants analysed through comprehensive phenotyping pipelines demonstrate a high level of pleiotropy[26, 27]. Indeed for the 67 genes uncovered via our screen, we find that the majority show additional phenotypes. This is true

the encoded proteins of the 52 genes with known human hereditary hearing loss dominant (DFNA), recessive (DFNB) and X-linked (DFNX) gene products. The predicted network interaction map shows that a significant number of known genes (65) formed a densely connected hub that incorporated 11 IMPC novel genes (Fig. 3). However, the majority of the 52 genes (41/52) were unconnected, free nodes, highlighting the unexplored and diverse

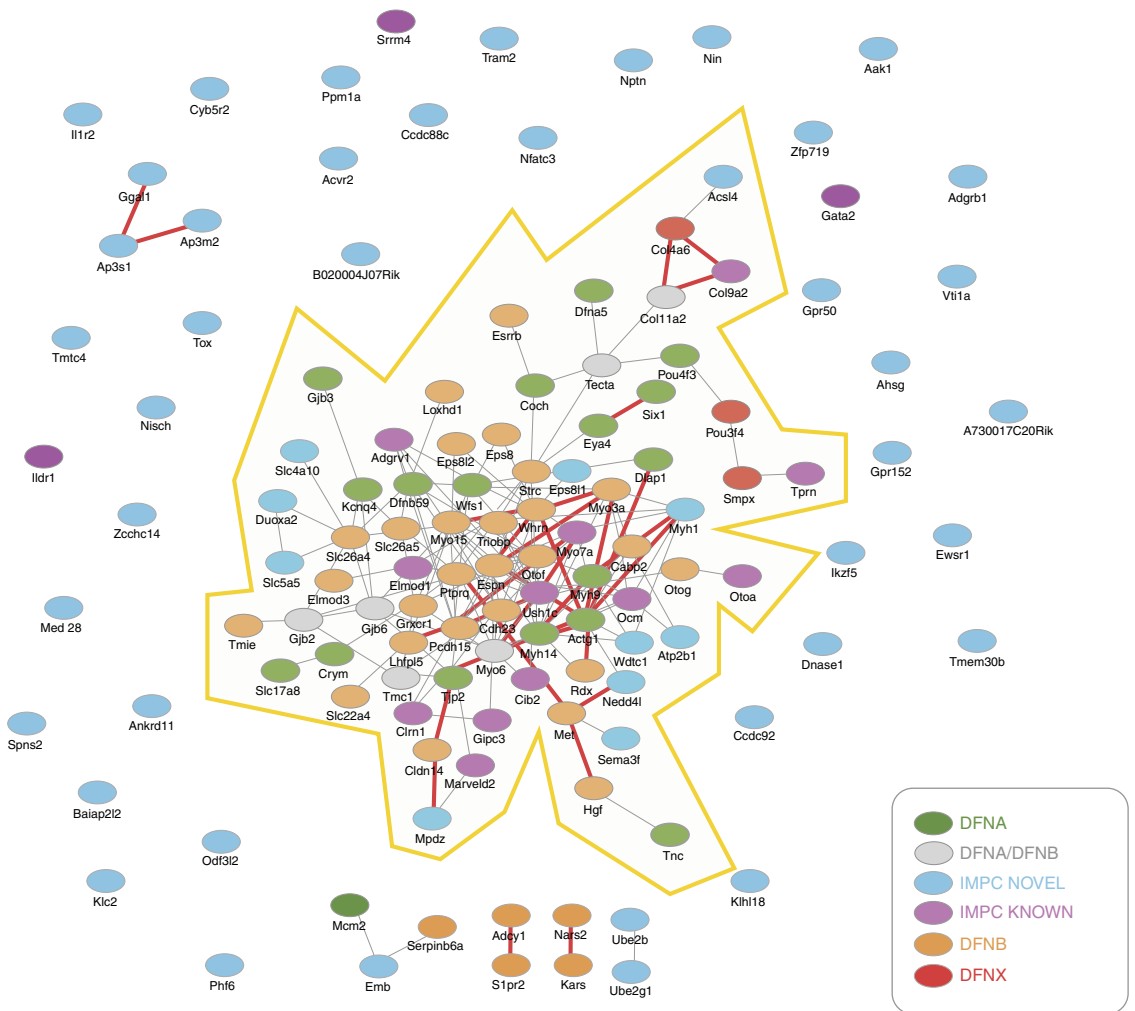

**Fig. 3** A network interaction map incorporating the proteins encoded by the 67 hearing loss genes identified from the IMPC ABR test. A STRING interaction map, incorporating known and predicted interactions, was generated for the known (15) and novel (52) IMPC hearing loss genes. *Blue nodes* are the IMPC novel candidate genes. Other nodes are previously reported genes that underlie hereditary hearing loss in humans, including DFNA (dominant hearing loss genes), DFNA/DFNAB (hearing loss genes showing both dominant and recessive inheritance), DFNB (recessive hearing loss genes), and DFNX (X-linked hearing loss genes). IMPC known genes are a subset of these genes and are highlighted by *purple colour*. The highly connected interaction map (*shaded*) consists of 65 known hearing loss genes and 11 novel candidate IMPC genes. The majority of IMPC novel candidate genes (41/52) are unconnected to the central network. *Thin grey edges* show interactions with a combined confidence score of ≥0.4, summated from evidence types: 'curated databases'; 'experimentally determined'; and, 'automated text mining'. *Bold red edges* show 'known' interactions with a combined confidence score of ≥0.4, summated from 'curated databases' and 'experimentally determined' only

for both known hearing loss genes, classified as non-syndromic in humans, as well as the novel candidate hearing loss genes. We cannot conclude from these data that the majority of mutants should be classified as syndromic, but rather as with many genetic and disease systems, hearing loss genes demonstrate considerable pleiotropy, which in some cases in humans may present as syndromic disease features. Finally, we could not discern any patterns or prevalence in terms of organ, disease or system areas across our gene set. This might require a more extensive data set covering the entire catalogue of loci involved in auditory function.

In our screen for auditory-related genes, we have explored around 15% of the mouse genome. Thus, we calculate that at a minimum the mammalian genome carries around 450 genes required for hearing function. In humans, around 150 non-syndromic hearing loss (DFNA, DFNB and DFNX) loci have been mapped, with 103 underlying genes identified (www.hereditaryhearingloss.org). There are over 400 genetic syndromes that include a hearing loss component[1, 2]. However, most of the genes underlying syndromic hearing loss are

unknown[5]. Our analysis thus reveals a very extensive and unexplored genetic landscape of genes required for auditory function and uncovers a large set of novel candidate hearing loss genes. We should emphasise that 450 genes represents a minimum set, as our screen may have failed to uncover several classes of genes, which contribute to hearing in either mouse or human. First, we discuss above our estimate of the false omission rate arising from our stringent manual curation of the set of 328 candidate genes. Moreover, we cannot rule out that there are hearing loss genes, in particular those with modest threshold effects, which our ABR test failed to identify. Also our analysis of the ABR data from 3006 genes did not include a waveform analysis that may have revealed further mutants. Second, we are unable to assess the contribution to auditory function of embryonic lethal, developmental mutations, which on different genetic backgrounds or in different genetic contexts may be homozygous viable and manifest a hearing loss phenotype not detectable in heterozygotes. Third, the hearing loss screen will not identify age-related hearing loss (presbycusis) genes, which again are likely to add

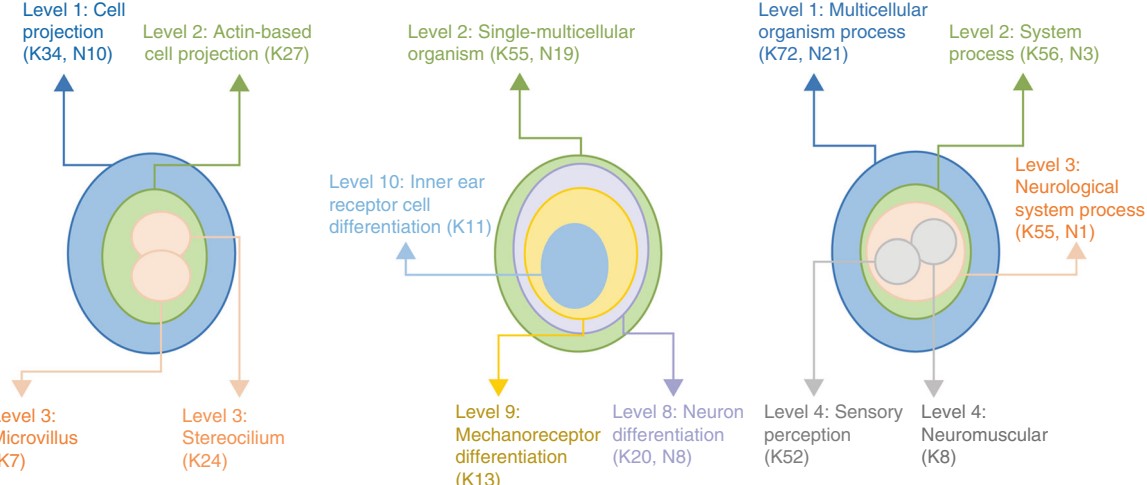

**Fig. 4** Distribution of IMPC known and IMPC novel hearing loss genes on three different gene ontology-directed acyclic graphs. Over-represented gene ontology terms for a joint data set of IMPC known (K) and IMPC novel (N) hearing loss genes were identified using gProfiler (Supplementary Table 4). The number of genes associated with any given term were counted and split into known and novel groups. Manual inspection of the enriched gene ontology terms and the gene count highlighted examples, where IMPC novel candidate genes were depleted at the lower ontology level. Here we show three examples of enriched gene ontology terms mapped onto their respective directed acyclic graphs (DAGs), alongside the number of known and novel candidate genes associated with enriched terms. For instance, the first example shows 34 known genes and 10 IMPC novel genes associated with the level 1 GO term 'Cell Projection'. From the 34 genes, 24 are annotated with 'Stereocilium' at level 3 while no novel candidate genes are annotated at this level

substantially to the total number of hearing loss loci[20]. Finally, the data from a parallel screen of around 1000 genes reports similar findings to IMPC[27].

In summary, the IMPC programme has identified a significant number of novel candidate hearing loss genes, amplifying our understanding of the auditory genetic landscape and providing the bases for new insights into auditory mechanisms, as well as a fund of new candidate genes for the many unidentified loci in the human population. IMPC continues to screen mutants for hearing loss phenotypes and expects the data set of novel genes to grow further over the coming years.

## Methods

**Ethical approval.** IMPC Centres breeding mice and collecting phenotyping data were guided by their own ethical review panels and licensing and accrediting bodies, reflecting the national legislation under which they operate. Details of their ethical review bodies and licenses are provided in Table 2. All efforts were made to minimise suffering by considerate housing and husbandry. All phenotyping procedures were examined for potential refinements that were disseminated throughout the Consortium. Animal welfare was assessed routinely for all mice involved.

**Mouse generation.** In the majority of cases, targeted ES cell clones were obtained from the European Conditional Mouse Mutagenesis Program (EUCOMM) and Knockout Mouse Project (KOMP) resource[12, 28] and injected into BALB/cAnN or C57BL/6J blastocysts for chimera generation. The resulting chimeras were mated to C57BL/6N mice, and the progeny were screened to confirm germline transmission. In a small number of cases, CRISPR/Cas9-mediated non-homologous end joining was utilised to generate small loss-of-function indels via pronuclear injection of C57BL/6N zygotes. In both instances, following the recovery of germline-transmitting progeny, heterozygotes were intercrossed to produce homozygous mutants. All strains are available from http://www.mousephenotype.org/.

**Genotyping and allele quality control.** Each mutant line underwent allele validation at the centre where the mouse model was produced. The consortium website (http://www.mousephenotype.org) has a webpage dedicated to each gene with a link to the allele created. Targeted alleles were validated by a combination of short-range PCR, quantitative PCR and non-radioactive Southern blot[29, 30]. These included allele-specific assays undertaken at the level of the ES cell and of the resulting mouse. The CRISPR-engineered alleles were validated by PCR amplification of the targeted locus and Sanger sequencing to confirm introduction of the required editing event. All QC data, including for CRISPR-engineered alleles and the guides employed, and Sanger sequencing of the engineered allele, are centrally deposited in the iMits (International Micro-injection tracking system) database.

This centrally held IMPC database tracks allele generation and enables all mouse clinics to assess data of any generated strain. These data are freely available on request. In addition, when mice or germplasms are distributed from an IMPC repository, the full allele-specific genotyping details are communicated.

**Housing and Husbandry.** Housing and husbandry data are captured for each IMPC Centre as described in Karp et al.[31] and is available from the IMPC portal (http://www.mousephenotype.org/about-impc/arrive-guidelines). In addition, pertinent metadata parameters for each IMPC centre are also available from the portal including, cage-type, caging density, bedding and enrichment details, feed constituents, lighting regimes, temperature and humidity of animal holding rooms, and full strain nomenclature.

**Auditory threshold phenotyping data collection.** We have used data collected from the IMPC high-throughput phenotyping pipeline, which is based on a pipeline design where a mouse is characterised by a series of standardised and validated set of tests underpinned by standard operating procedures (SOPs). The IMPReSS (International Mouse Phenotyping Resource of Standardised Screens) database (https://www.mousephenotype.org/impress), defines the experimental design, detailed procedural information, the data that is to be collected, age of the mice, significant metadata parameters and data QC for IMPC pipeline tests. Unlike most experiments, we cannot randomly allocate animals to experiment groups, rather we rely on Mendelian inheritance to provide the randomisation method. However, at different institutes a variety of approaches are taken to minimise bias such as order effects including: alternate animal order; cage casual randomisation; and casual randomisation within a cage. There were no consistent approaches to blinding for ABR data collection and annotation across the institutes within IMPC. These issues are discussed with regard to the ARRIVE guidelines at http://www.mousephenotype.org/about-impc/arrive-guidelines.

The IMPC phenotyping pipeline included an ABR test[32], carried out at 14 weeks, to determine hearing sensitivity using evoked potential recordings in anaesthetised mice (IMPC_ABR_002). Hearing was assessed at five frequencies—6, 12, 18, 24 and 30 kHz. Tone pips were presented from 0 to 85 dB sound pressure level (SPL) in 5 dB intervals and were 5 ms in duration with a 1 ms rise/fall, presented 256 times at 42.6/sec. If hearing loss was suspected for a particular mutant line (e.g., absence of ABR waveforms at any stimulus level), stimulus presentation levels were extended to 95 dB SPL. Tone stimuli were presented in decreasing frequency order for a particular sound level and from low to high stimulus level. A click-evoked ABR is optional, and some centres, but not all, also acquired click data, but we have not included this in our analysis. The ABR test is conducted open field with hearing thresholds determined for one ear. Importantly, each centre records significant metadata parameters including: equipment manufacturer; equipment model; recording environment; anaesthetic agent; and anaesthetic dose. In addition, the Combined SHIRPA and Dysmorphology test [IMPC_CSD_003], carried out at 9 weeks, includes a click box test that provides a crude assessment of hearing[9]. The click box, which emits a high-frequency (~20 kHz)

**Table 2 Ethical review board information for each phenotyping centre**

| Institute | Information |
|---|---|
| BCM Baylor College of Medicine | Approval committee: Institutional Animal Care and Usage Committee |
| | Approval Licence: AN-5896 |
| GMC Helmholtz Zentrum München | Approval committee: Regierung von Oberbayern |
| | Approval Licence: 2532 |
| ICS Mouse Clinical Institute | Approval Committee: Com'Eth. agreement nb: 17 |
| | Approval licences: internal numbers 2012-009 and 2014-024. |
| | Approval from the Ministry of research: APAFIS#4789-20J6040511578546v2 |
| MRC Harwell | Approval committee: Animal Welfare and Ethical Review Body (AWERB) |
| | Approval Licence: 30/2890 |
| Nanjing University | Approval committee: IACUC of MARC |
| | Approval Licence: NRCMM9 |
| RBRC RIKEN Tsukuba Institute, BioResource Center | Approval committee: The RIKEN Tsukuba Animal Experiments Committee |
| | Approval Licence: Approval Number: Exp14—010 Research title: Collection, maintenance, storage, breeding and distribution of the mouse strains for the Biological Resource |
| The Jackson Laboratory | Approval committee: The Jackson Laboratory Institutional Animal Care and Use Committee (IACUC) |
| | Approval Licence: Institutional Permit: NIH Office of Laboratory |
| | Animal Welfare (OLAW) # A3268-01 OLAW |
| | Assurance # 811101 Production Grant IACUC |
| | Protocol: 99066 Phenotyping grant Animal Use |
| | Summary IACUC protocol 11005 |
| The Centre for Phenogenomics | Approval committee: Animal Care Committee (ACC) of The Centre for Phenogenomics |
| | Approval Licence: Animal Use Protocol (AUP) 0153, 0275, 0277, 0279 |
| UCD University of California, Davis | Approval committee: UC Davis Institutional Animal Care and Use Committee (IACUC) |
| | Approval Licence: Protocol #18119 |
| WTSI Wellcome Trust Sanger Institute | Approval committee: Animal Welfare and Ethical Review Body (AWERB) |
| | Approval Licence: PPL 80/2076 Valid 27 November 2006—3 January 2012; PPL 80/2485 valid 22 December 2011—3 January 2017 |

tone stimulus at 90 dB SPL, was held 30 cm above the mouse. When the click box is activated, a hearing mouse will elicit a 'Preyer' reflex (the pinnae flick backwards). A lack or reduction of Preyer reflex indicates possible hearing impairment.

**ABR experimental design**. At each IMPC Centre, ABR phenotyping data from each mutant strain, and age-matched wild-type mice of equivalent genetic backgrounds, were collected at 14 weeks of age. When possible, cohorts of at least four mutant mice were tested, with a preference for two male and two female mice. If no homozygotes were obtained from 28 or more offspring of heterozygote intercrosses, the strain was scored non-viable. Similarly, if <13% of the pups resulting from intercrossing were homozygous, the line was scored as being subviable. In such circumstances, heterozygote mice were used in the phenotyping pipelines. The random allocation of mice to experimental group (wild-type vs. knockout) was driven by Mendelian inheritance. The individual mouse was considered the experimental unit within the studies. Further detailed experimental design information (e.g., exact definition of a control animal) for each phenotyping Centre, or the blinding strategy implemented is captured with a standardised ontology as detailed in Karp et al.[31] and is available from the IMPC portal (http://www.mousephenotype.org/about-impc/arrive-guidelines). As a high-throughput project, the target sample size of four animals per knockout strain is relatively low. The IMPC determined to set the lowest sample size that would consume the least amount of resources while achieving the goal of detecting auditory threshold abnormalities. At times, viability issues or the difficulty in administering the ABR test might further limit the number of animals. As such, when data are visualised on the IMPC portal, the number of animals phenotyped is listed.

**Data quality control**. The data generated in each IMPC centre are captured centrally by the Data Coordination Centre (DCC), where a team of data wranglers perform QC. The QC process involves data wranglers checking all data both manually and with automated methods. QC issues are raised through a QC web interface, where IMPC centres can respond to confirm it is an issue or alternatively that the data are correct. Each QC issue is then tracked with the corresponding data points until it has been corrected by the data contributing centre. Once the data have passed QC, it is released to the Core Data Archive (CDA) at the European Bioinformatics Institute, through a regular release schedule. The data set analysed here consists largely of data from the IMPC data release 5.0 with a smaller set of pre-QC data from the DCC. For the purpose of this article, the ABR data in this smaller set were QC'd and manually curated.

**Wild-type data sets**. Wild-type data sets were assembled for a trait by selecting wild-type mice that were generated by the same IMPC Centre. The wild-type mice are analysed with the same genetic background, sex, pipeline and metadata parameters (e.g., instrument).

**Statistical analysis**. ABR data from IMPC release v5.0 along with additional pre-QC data deposited up until 6 September 2016 was used for the detection of putative pheno-deviants. In total, data from 3006 mutant strains were available for analysis. Each knockout strain was examined in three ways: only males, only females and both sexes combined. A minimum of 20 wild-type data points and 3 mutant data points was required for the analysis to proceed. The parameters assigned for annotation in the ABR procedure, stored in IMPReSS (http://www.mousephenotype.org/impress/) were analysed by comparing the knockout strain with the appropriate wild-type control data. The default statistical analysis employed for the IMPC annotation pipeline is a mixed model approach, or a Mann–Whitney inference test may be employed as an alternative. However, for analysis of ABR data neither approach is suitable when there is lack of variation within the data set (e.g., if all mutant mice are profoundly deaf at all the frequencies tested) or due to the step nature of the data, respectively. Therefore for this analysis, we used an enhanced version of PhenStat version 2.6.0[13] that includes a reference range plus approach, which allows identification of hearing impairment even when there are few data points or a lack of variation within the data set. The 'testDataset' function within range plus approach was used with a 98% reference range, produced from the wildtypes for each sex. Per knockout line, range plus approach classifies phenotype values as normal or low (below the 1% critical value) or high (above the 99% critical value) for each sex. A Fisher's exact test is used to compare the proportions of classification, independent of sex, between wild-type and knockout (low/normal vs. high and high/normal vs. low) and the lowest $p$-value returned. This approach was used due to the small sample size of the experiment. A critical $p$-value ($p \leq 0.01$) was used to generate a list of putative auditory pheno-deviant knockout lines. This list was used for further manual curation by domain experts.

**Assessing the functional enrichment of hearing loss genes**. A single data set consisting of 103 known human hereditary hearing loss genes (DFNA, DFNB and DFNX) and 52 IMPC novel hearing loss genes was used for functional assessment. We used g:Profiler (version r1665_e85_eg32) to find enrichment within the gene ontology, categories include; molecular function, cellular component and biological process[33]. Results were corrected for multiple testing using g:Profiler's internal method g:SCS, with enrichments considered significant at $q < 0.05$. Assessment of

protein interactions was undertaken using STRING (version 10.0) at a combined medium confidence setting of ≥0.4[34].

**Data availability**. Data from this study are available from IMPC at www.mousephenotype.org. An API in the form of RESTful interfaces is available for accessing post-QC data from the Core Data Archive (http://www.mousephenotype.org/data/documentation/data-access), and pre-QC data are also available for manual download (via Phenoview) or on request (http://www.mousephenotype.org/contact-us).

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

## Acknowledgements

Research reported in this publication was supported by: Agence Nationale de la Recherche (ANR-10-IDEX-0002-02, ANR-10-INBS- 07 PHENOMIN to Y.H.); Centre Européen de Recherche en Biologie et en Médecine (Y.H.); European Commission (EUMODIC contract no. LSHG-CT-2006-037918 to G.P.T.-V, W.W., M.H.d.A., K.L.S., K.P.S. and S.D.M.B.); French National Institute of Health and Medical Research—INSERM (Y.H.); Government of Canada through Genome Canada and Ontario Genomics (OGI-051 to C.M. and S.D.M.B.); Medical Research Council (53650 to S.W. and S.D.M.B.) (G0300212 and MC_qA137918 to K.P.S.) (MC_U142684175 and MC_U142684172 to S.D.M.B.); National Centre for Scientific Research—CNRS (Y.H.); National Institutes of Health (U42OD011185 to S.A.M.) (UM1OD023222 to S.A.M., K.L.S. and R.E.B.) (U54HG006332 to K.L.S. and R.E.B.) (U42OD011175, U54HG006364, 3U54HG006364-03S1 and UM1OD023221 to K.C.K.L. and C.M.) (U54HG006348 and 2UM1HG006348 to A.B., S.W. and S.D.M.B.) (U54HG006370 and 2UM1HG006370 to W.C.S., T.F.M., P.F., A.-M.M., H.E.P., D.S. and S.D.M.B.); Wellcome Trust (098051 and 100699 to K.P.S.). The content is solely the responsibility of the authors and does not necessarily represent the official views of the National Institutes of Health.

## Author contributions

M.R.B., K.P.S. and S.D.M.B.: Conceived and directed the study. N.J.I., G.P.T.-V., X.G., A.B., W.C.S., M.M., A.L.B., M.J.J., J.S., M.E.D., W.W., M.H.d.A, Y.H., S.W., L.M.J.N., A. M.F., C.M., S.A.M., K.L.S., R.E.B., D.B.W., K.C.K.L., D.J.A., J.W., N.Ka., P.F., D.S., T.F.M., H.E.P., L.M.T., S.W., K.P.S., A.-M.M. and S.D.M.B.: Directed the research at their respective institutes. M.R.B., M.M.S., N.J.I., S.G., L.S., H.C., S.T., J.M., N.Ku., S.P., L.R.B., D.A.C., H.M., P.R., O.M., L.K., N.Ka., P.F., D.S., T.F.M., H.E.P. and A.-M.M.: Generated data, developed data tools and databases, and/or carried out data and statistical analyses. M.R.B. and S.D.M.B.: Wrote the manuscript

## Additional information

**Competing interests**: The authors declare no competing financial interests.

**Publisher's note**: 

Michael R. Bowl[1], Michelle M. Simon[1], Neil J. Ingham[2,3], Simon Greenaway[1], Luis Santos[1], Heather Cater[1], Sarah Taylor[1], Jeremy Mason[4], Natalja Kurbatova[4], Selina Pearson[3], Lynette R. Bower[5], Dave A. Clary[5], Hamid Meziane[6], Patrick Reilly[6], Osamu Minowa[7], Lois Kelsey[8,9,10], The International Mouse Phenotyping Consortium, Glauco P. Tocchini-Valentini[11], Xiang Gao[12], Allan Bradley[3], William C. Skarnes[3], Mark Moore[13], Arthur L. Beaudet[14], Monica J. Justice[8,9,10,14], John Seavitt[14], Mary E. Dickinson[15], Wolfgang Wurst[16], Martin Hrabe de Angelis[17], Yann Herault[6,18,19,20], Shigeharu Wakana[7], Lauryl M.J. Nutter[8,9,10], Ann M. Flenniken[8,9,10], Colin McKerlie[8,9,10], Stephen A. Murray[21], Karen L. Svenson[21], Robert E. Braun[21], David B. West[22], K.C. Kent Lloyd[5], David J. Adams[3], Jacqui White[3], Natasha Karp[3], Paul Flicek[4], Damian Smedley[23], Terrence F. Meehan[4], Helen E. Parkinson[4], Lydia M. Teboul[1], Sara Wells[1], Karen P. Steel[2,3], Ann-Marie Mallon[1] & Steve D.M. Brown[1]

[1]Medical Research Council Harwell Institute (Mammalian Genetics Unit and Mary Lyon Centre), Harwell, Oxfordshire OX11 0RD, UK. [2]King's College London, London SE1 1UL, UK. [3]The Wellcome Trust Sanger Institute, Wellcome Trust Genome Campus, Hinxton, Cambridge CB10 1SA, UK. [4]European Molecular Biology Laboratory, European Bioinformatics Institute, Wellcome Trust Genome Campus, Hinxton, Cambridge CB10 1 SD, UK. [5]Mouse Biology Program, University of California, Davis, California 95618, USA. [6]CELPHEDIA, PHENOMIN, Institut Clinique de la Souris (ICS), 1 rue Laurent Fries, Illkirch-Graffenstaden F-67404, France. [7]RIKEN BioResource Center, Tsukuba, Ibaraki 305-0074, Japan. [8]The Centre for Phenogenomics, Toronto, Ontario, Canada M5T 3H7. [9]The Hospital for Sick Children, Toronto, Ontario, Canada M5G 1X8. [10]Canada and Mount Sinai Hospital, Toronto, Ontario, Canada M5G 1X5. [11]Monterotondo Mouse Clinic, Italian National Research Council (CNR), Institute of Cell Biology and Neurobiology, I-00015 Monterotondo Scalo, Italy. [12]SKL of Pharmaceutical Biotechnology and Model Animal Research Center, Collaborative Innovation Center for Genetics and Development, Nanjing Biomedical Research Institute, Nanjing University, 210061 Nanjing, China. [13]IMPC, San Anselmo, California 94960, USA. [14]Department of Molecular and Human Genetics, Baylor College of Medicine, Houston, Texas 77030, USA. [15]Department of Molecular Physiology and Biophysics, Baylor College of Medicine, Houston, Texas 77030, USA. [16]Institute of Developmental Genetics, Helmholtz Zentrum München, German Research Center for Environmental Health GmbH, Ingolstaedter Landstrasse 1, 85764 Neuherberg, Germany. [17]German Mouse Clinic, Institute of Experimental Genetics, Helmholtz Zentrum München, German Research Center for Environmental Health GmbH, Ingolstaedter Landstrasse 1, 85764 Neuherberg, Germany. [18]Institut de Génétique et de Biologie Moléculaire et Cellulaire (IGBMC), Université de Strasbourg, 67404 Illkirch, France. [19]Centre National de la Recherche Scientifique, UMR7104, 67404 Illkirch, France. [20]Institut National de la Santé et de la Recherche Médicale, U964, 67404 Illkirch, France. [21]The Jackson Laboratory, Bar Harbor, Maine 04609, USA. [22]Childrens' Hospital Oakland Research Institute, Oakland, California 94609, USA. [23]Queen Mary University of London, London WC1E 6BT, UK. Michael R. Bowl, Michelle M. Simon, Neil J. Ingham, Simon Greenaway and Luis Santos contributed equally to this work. Karen P. Steel, Ann-Marie Mallon and Steve D. M. Brown jointly supervised this work
A full list of the consortium members appears below

## The International Mouse Phenotyping Consortium

Sue Allen[1], Sharon Clementson-Mobbs[1], Gemma Codner[1], Martin Fray[1], Wendy Gardiner[1], Russell Joynson[1], Janet Kenyon[1], Jorik Loeffler[1], Barbara Nell[1], Andrew Parker[1], Deen Quwailid[1], Michelle Stewart[1], Alison Walling[1], Rumana Zaman[1], Chao-Kung Chen[4], Nathalie Conte[4], Peter Matthews[4], Mike Relac[4], Ilinca Tudose[4], Jonathan Warren[4], Elise Le Marchand[6], Amal El Amri[6], Leila El Fertak[6], Hamid Ennah[6], Dalila Ali-Hadji[6], Abdel Ayadi[6], Marie Wattenhofer-Donze[6], David Moulaert[6], Sylvie Jacquot[6], Philippe André[6], Marie-Christine Birling[6], Guillaume Pavlovic[6], Valérie Lalanne[6], Aline Lux[6], Fabrice Riet[6], Christophe Mittelhaeuser[6], Raphael Bour[6], Alain Guimond[6], Chaouki Bam'Hamed[6], Sophie Leblanc[6], Laurent Vasseur[6], Mohammed Selloum[6], Tania Sorg[6], Shinya Ayabe[7], Tamio Furuse[7], Hideki Kaneda[7], Kimio Kobayashi[7], Hiroshi Masuya[7], Ikuo Miura[7], Yuichi Obata[7], Tomohiro Suzuki[7], Masaru Tamura[7], Nobuhiko Tanaka[7], Ikuko Yamada[7], Atsushi Yoshiki[7], Zorana Berberovic[8,9,10], Mohammed Bubshait[8,9,10], Jorge Cabezas[8,9,10], Tracy Carroll[8,9,10], Greg Clark[8,9,10], Shannon Clarke[8,9,10], Amie Creighton[8,9,10], Ozge Danisment[8,9,10], Mohammad Eskandarian[8,9,10], Patricia Feugas[8,9,10], Marina Gertsenstein[8,9,10], Ruolin Guo[8,9,10], Jane Hunter[8,9,10], Elsa Jacob[8,9,10], Qing Lan[8,9,10], Valerie Laurin[8,9,10], Napoleon Law[8,9,10], Sue MacMaster[8,9,10], David Miller[8,9,10], Lily Morikawa[8,9,10], Susan Newbigging[8,9,10], Celeste Owen[8,9,10], Patricia Penton[8,9,10], Monica Pereira[8,9,10], Dawei Qu[8,9,10], Xueyuan Shang[8,9,10], Gillian Sleep[8,9,10], Khondoker Sohel[8,9,10], Sandra Tondat[8,9,10],

Yanchun Wang[8,9,10], Igor Vukobradovic[8,9,10], Yingchun Zhu[8,9,10], Francesco Chiani[11], Chiara Di Pietro[11], Gianfranco Di Segni[11], Olga Ermakova[11], Filomena Ferrara[11], Paolo Fruscoloni[11], Aalessia Gambadoro[11], Serena Gastaldi[11], Elisabetta Golini[11], Gina La Sala[11], Silvia Mandillo[11], Daniela Marazziti[11], Marzia Massimi[11], Rafaele Matteoni[11], Tiziana Orsini[11], Miriam Pasquini[11], Marcello Raspa[11], Aline Rauch[11], Gianfranco Rossi[11], Nicoletta Rossi[11], Sabrina Putti[11], Ferdinando Scavizzi[11], Giuseppe D. Tocchini-Valentini[11], Joachim Beig[16], Antje Bürger[16], Florian Giesert[16], Jochen Graw[16], Ralf Kühn[16], Oskar Oritz[16], Joel Schick[16], Claudia Seisenberger[16], Oana Amarie[16,17], Lillian Garrett[16,17], Sabine M. Hölter[16,17], Annemarie Zimprich[16,17], Antonio Aguilar-Pimentel[17], Johannes Beckers[17], Robert Brommage[17], Julia Calzada-Wack[17], Helmut Fuchs[17], Valérie Gailus-Durner[17], Christoph Lengger[17], Stefanie Leuchtenberger[17], Holger Maier[17], Susan Marschall[17], Kristin Moreth[17], Frauke Neff[17], Manuela A. Östereicher[17], Jan Rozman[17], Ralph Steinkamp[17], Claudia Stoeger[17], Irina Treise[17], Tobias Stoeger[17,24,25], Ali Önder Yildrim[17,24,25], Oliver Eickelberg[24,25], Lore Becker[17,26], Thomas Klopstock[26], Markus Ollert[27,28], Dirk H. Busch[29], Carsten Schmidt-Weber[30], Raffi Bekeredjian[31], Andreas Zimmer[32], Birgit Rathkolb[17,33], Eckhard Wolf[33], Martin Klingenspor[34]

[24]Comprehensive Pneumology Center, Institute of Lung Biology and Disease, Helmholtz Zentrum München, German Research Center for Environmental Health (GmbH), Ingolstädter Landstraße 1, 85764 Neuherberg, Germany. [25]German Center for Lung Research, Aulweg 130, 35392 Gießen, Germany. [26]Department of Neurology, Friedrich-Baur-Institut, Ludwig-Maximilians-Universität München, Ziemssenstrasse 1a, 80336 Munich, Germany. [27]Department of Infection and Immunity, Luxembourg Institute of Health, Esch-sur-Alzette L-4354, Luxembourg. [28]Department of Dermatology and Allergy Center, Odense Research Center for Anaphylaxis, University of Southern Denmark, Odense DK-5000, Denmark. [29]Institute for Medical Microbiology, Immunology and Hygiene, Technical University of Munich, Trogerstrasse 9, 81675 Munich, Germany. [30]Center of Allergy & Environment (ZAUM), Technische Universität München, and Helmholtz Zentrum München, Ingolstaedter Landstrasse, 85764 Neuherberg, Germany. [31]Department of Cardiology, University of Heidelberg, Im Neuenheimer Feld 410, 69120 Heidelberg, Germany. [32]Institute of Molecular Psychiatry, Medical Faculty, University of Bonn, Sigmund-Freud-Strasse 25, 53127 Bonn, Germany. [33]Ludwig-Maximilians-Universität München, Gene Center, Institute of Molecular Animal Breeding and Biotechnology, Feodor-Lynen Strasse 25, 81377 Munich, Germany. [34]Institute for Food and Health, Technical University Munich, Gregor-Mendel-Str. 2, 85350 Freising-Weihenstephan, Germany

