## [peer review file · Nature Communications]

Reviewer #1 (Remarks to the Author):

The authors report the results of the screen for hearing loss of 3006 knockout mouse strains generated by the International Mouse Phenotyping Consortium (IMPC). To identify the genes which mutation impact hearing, the authors aimed at recording the auditory brainstem response (ABR) thresholds at 5 frequencies (6, 12, 18, 24 and 30 kHz) in 4 14-week-old mutant animals carrying either homozygote mutations or if lethal, heterozygote mutation of each gene of interest.

Using this approach the authors identified 67 genes which expression seems necessary for hearing. 52 of these genes were not known to be crucial for hearing before this study.

Comparison of the ABR of these mutants at different frequencies led to their classification into genes associated with severe, or mild hearing loss at all frequencies, or high or low frequency hearing loss. Further computational studies including GO analysis did not identify a common pathway these genes are part of, but rather pointed at the diversity of functions they may be involved in.

The rationale for this study is strong, identification of potential genes which mutation leads to hearing loss is extremely useful to the community.

Clarity of the manuscript could be improved in several places, and an appropriate level of caution in the conclusions added. The methods although crucial for judging of the strength of the conclusions of this report are only mostly described in supplementary data. The manuscript would gain in having these methods more clearly explained and presented in the core of the manuscript.

The authors conclude at the identification of 52 genes crucial for hearing. I am not sure the methods support such a strong conclusion, this work rather point at “candidate genes” potentially involved in hearing. Sample size is very small and statistical analyses for ABR comparison is unusual.

Major comments:

1. Please indicate whether the ABR were done on the right ear, the left ear, or both. Are we looking at the results of the best ear or the average of both ears?
2. Before concluding that the gene mutated is directly involved in hearing, presence of an otitis media (inflammation of the middle ear) should be excluded. Which test(s) were done to ascertain this?

3. Nothing in the manuscript suggests that the potential hearing loss detected here is isolated and not associated with other symptoms, and as such “syndromic”.

Were other phenotype(s) identified in mice carrying a mutation of the gene candidates? If so, please specify in the manuscript. This may allow for a better interpretation of the results or their limitations.

As an example, patients with craniofacial abnormalities (such as Down syndrome, cri du chat syndrome, cleft lip and/or cleft palate, velocardiofacial syndrome) have an increased risk in developing otitis media.

4. Statistical analysis whether detailed in the core of the manuscript or in the supplementary data need further explanation.

Please explain better the choice of the complex and unusual statistical analysis used here to reach a conclusion, as it differs significantly from standard tests used in the auditory field. Why choose a Fisher’s exact test to compare quantitative results?

P4, lines 25: please further explain how the reference range was calculated and why using this approach rather than the standard tests recognized in the field?

P4, line 31 “we determined a suitable reference range and critical p value by the examination of known deafness genes....” Please further explain.

How much real knowledge can we gain from a software that allows “identification of ABR related hearing loss when there are insufficient data points...” p.13 lines 4 to 13.

5. Please explain more clearly the bases of your manual curation p.37 line 38 is stated “we eliminated genes where a single mid-range frequency showed a significant difference...”

Isn’t the lack of reproducibility of the results between animals carrying the same mutation an essential criterion of elimination? This is only indicated in the supplementary data and should be in the core of the manuscript. Identification of potential differences depending on the sex of the animal is also a very interesting result which should be found in the core of the manuscript rather than just in the supplementary data. Yet can a conclusion be reached with n=2? This is doubtful.

6. Please rewrite abstract so it includes an actual summary of the results and methods used. Considering the limited body of proof presented, please refrain from using “deafness genes” along the manuscript and replace by “candidates genes”.

Other comments:

1. Please consider changing “deafness” along the manuscript for “hearing loss” when more appropriate.
2. Please change the first sentence of the abstract p. 4, line 1, it is not clear.
3. Please include in the introduction a brief description of how the null mutants were generated.
4. Measurement of ABR to a broadband click stimulus is mentioned in the introduction and the legend of several figures but the results are not presented, please either include these results or discuss why they were not included.
5. ABR waveform analysis: was there any abnormalities in the waveform shape recorded for the different mutant mice? Any wave I latency change? Any marked and reproducible amplitude change?
6. Please define “QC” when first used p. 4, line 19
7. Here the results identify 52 new genes and 15 known genes. It is not clear to me that emphasizing that only 9 out of 261 genes excluded are known deafness genes is a good criterion: by comparison with the results presented here we could “miss” a large number of new genes.
8. P. 5 line 20 please correct to “The products of the 52 novel deafness genes...”
9. P. 5 line 11: precise which criteria were used to classify the genes in these 4 classes of hearing loss in the core of the manuscript.
10. P. 6 lines 10 to 25 This discussion should include the estimation of the number of genes implicated in syndromic deafness (which is at least 400). Those may be a better reference for comparison with the results obtained here.
11. For a long time, the presence of vestibular defects easily visualized (circling behavior...) prompted the test for potential hearing deficit in mutant mice. Do you have any data regarding possible vestibular defects in the mutant mice presented here? Please include data if available or comment.
12. Some of the “new” genes identified, for example *Atp2b1* (other name *PMCA1*) or *Sema3f*, are already known to have key function in the inner ear. Please discuss and reference appropriately.

Supplementary data:

p.11 line 22: define SOPs

p.12 line 25: use “article” instead of paper

p.12 line 30: please correct or explain what you mean by “zygosity (either homozygous or heterozygous)” in this context.

Please indicate how the mice were anesthetized before ABRs.

Supplementary Table 4. Please add for each comparison the number of animals or “ears” included in the “Comparison”.

Reviewer #2 (Remarks to the Author):

This unusual submission from Bowl et al. describes a large-scale screen for deafness in many lines (>3000) that have been generated by the International Mouse Phenotyping Consortium. They used very stringent criteria to ensure that they identified lines that truly had hearing loss, resulting in a total of 67 lines. Remarkably, 52 of those genes had never before been associated with deafness. They then used these values to extrapolate the total number of deafness genes in the mouse to be ~450, which is consistent with previous estimates that were based on considerably less reliable data.

This manuscript is an extremely valuable contribution. While the data are also freely searchable on the IMPC webpage, there was an extraordinary amount of work that went into the data generation. I have very few comments:

Page 5, line 4. There are two publications on neuroplastin in the inner ear. Please cite both.

Page 7, line 7. Remove “manuscript to be submitted.”

Fig. 1. The labels for the different centers should use the same font size as the other labels in the figure. They are very difficult to read. Do the authors wish to comment on the variability? Is this variability due to different instrumentation at each center?

Suppl. Fig. 2. Looks pretty but does it really say much? The problem with inner ear genes is that many of them are not widely expressed and have not been studied before. There simply is a lack of information.

Reviewer #1

The rationale for this study is strong, identification of potential genes which mutation leads to hearing loss is extremely useful to the community.

We would like to thank the reviewer for their supportive comments.

Major comments

1) *Please indicate whether the ABR were done on the right ear, the left ear, or both. Are we looking at the results of the best ear or the average of both ears?*

The results are from one ear tested in an open field set-up. We have added the following text (p.14, line 6):

The ABR test is conducted open field with hearing thresholds determined for one ear.

2) *Before concluding that the gene mutated is directly involved in hearing, presence of an otitis media (inflammation of the middle ear) should be excluded. Which test(s) were done to ascertain this?*

The aim of the programme was to capture all forms of hearing impairment, from conductive to sensorineural, in order to assess all of the key underlying genetic pathways for deafness. We believe it is important to uncover the genetic landscape that impacts on all forms of hearing impairment, and we amplify this thinking in a new section in the paper on p.8, line 22:

The screen we have undertaken is effectively unbiased for both the developmental and physiological origin of the auditory phenotype, as well as for syndromic features that might be associated with the observed deafness. Without further investigation of individual mutants, we are unable to distinguish between deafness due to a sensorineural origin in the inner ear, or due to conductive deafness due to middle ear problems, such as otitis media.

Moreover, we amplify these points in the discussion (p.11, line 12):

However, for the majority of novel candidate loci it will be important to further explore expression and function in the auditory apparatus. In particular, an ABR screen is relatively unbiased and will reveal auditory phenotypes that arise due to conductive hearing impairment as well as sensorineural hearing loss, or possibly a combination of both. We cannot rule out that some of the novel candidate deafness genes impact on middle ear function, or cause otitis media. This class of deafness genes is unlikely to be represented by mutations with severe hearing loss or high-frequency hearing loss, and would be more likely to be present in the class of genes with low-frequency or mild hearing loss. The mouse has proved an excellent model for the identification of genes involved with chronic otitis media^{23, 24, 25} and it will be interesting to determine if the novel set of candidate deafness genes uncovered here further extends our knowledge of the genetic pathways involved with this form of deafness.

3) *Nothing in the manuscript suggests that the potential hearing loss detected here is isolated and not associated with other symptoms, and as such "syndromic". Were*

other phenotype(s) identified in mice carrying a mutation of the gene candidates? If so, please specify in the manuscript. This may allow for a better interpretation of the results or their limitations. As an example, patients with craniofacial abnormalities (such as Down syndrome, cri du chat syndrome, cleft lip and/or cleft palate, velocardiofacial syndrome) have an increased risk in developing otitis media.

We have now added to the manuscript a detailed analysis of the additional phenotypes identified for each mutant following analysis through the IMPC phenotyping pipeline. These are summarised in Table 2 and we amplify the results and observations on p. 8-9. Sixty of the mutants show multiple phenotypes that reflects the extensive pleiotropy that is commonly observed for the vast majority of genes in the IMPC pipeline. Moreover, we have now catalogued the large number of mutants that show an absence of a Preyer reflex in the click box test as part of the Combined SHIRPA and Dysmorphology (CSD) test in the IMPC pipeline. The majority of mutants classified as *severe hearing loss* had an absent Preyer reflex (20 out of 25) – see p. 8 and Table 2. We also catalogued any observable vestibular dysfunction, and discuss this in response to point 11 in Other Comments (see below).

Overall, we document that apart from the CSD test, there is no obvious pattern to the prevalence of additional phenotypes for the mutants tested. We have now commented on the observed pleiotropy and the nature of non-syndromic vs syndromic deafness in the discussion (p.11, line 26):

The unbiased nature of the IMPC phenotyping pipeline encompassing a wide range of phenotyping tests covering various biological systems allowed us to explore the pleiotropic state of the mutants that we have discovered that may inform us about the syndromic nature of the candidate deafness genes we have identified. In the human population many forms of deafness present as a component of a wider genetic syndrome with a range of phenotypes. It is quite likely that some non-syndromic forms of deafness may also have additional unrecognised phenotypes. Large-scale studies of mutants analysed through comprehensive phenotyping pipelines demonstrate a high level of pleiotropy^{26, 27}. Indeed for the 67 genes uncovered via our screen, we find that the majority show additional phenotypes. This is true for both known deafness genes, classified as non-syndromic in humans, as well as the novel candidate deafness genes. We cannot conclude from these data that the majority of mutants should be classified as syndromic, but rather as with many genetic and disease systems, deafness genes demonstrate considerable pleiotropy which in some cases in humans may present as syndromic disease features.

4) *Statistical analysis whether detailed in the core of the manuscript or in the supplementary data need further explanation. Please explain better the choice of the complex and unusual statistical analysis used here to reach a conclusion, as it differs significantly from standard tests used in the auditory field. Why choose a Fisher's exact test to compare quantitative results?*

The default analysis used by the IMPC annotation pipeline is a mixed model approach. A Mann-Whitney inference test is employed as an alternative. However, reflecting the nature of ABR data we found some analyses failed to fit a Mixed-Model approach, for example due to a lack of variation within the dataset (e.g. all the knockout animals were profoundly deaf) and subsequently fails to be suitable for a Mann-Whitney inference test (due to the step nature of the data). In order to undertake a robust and consistent analysis, a third approach was required, namely the Reference Range Plus technique. Reference Range Plus is also part of the R 'PhenStat' package, but it is not used in the default analysis pipeline within the IMPC project.

When using a reference range (constructed from the wildtype data) a contingency table is created counting the number of measurements within and without the reference range. The optimum inferential test for such a 2x2 table is the Fisher's Exact Test. No conclusions were inferred from these results until a domain expert accessed each genotype line for genuine pheno-deviancy.

As such we have modified the text in the methods as follows (p.15, line 21):

...The parameters assigned for annotation in the ABR procedure, stored in IMPReSS (<http://www.mousephenotype.org/impress/>) were analysed by comparing the knockout strain with the appropriate wildtype control data. The default statistical analysis employed for the IMPC annotation pipeline is a mixed model approach, or a Mann-Whitney inference test may be employed as an alternative. However, for analysis of ABR data neither approach is suitable when there is lack of variation within the dataset (e.g. if all mutant mice are profoundly deaf at all the frequencies tested) or due to the step nature of the data, respectively. Therefore for this analysis we used an enhanced version of PhenStat version 2.6.0 that includes a reference range plus approach (RRP), which allows identification of hearing impairment even when there are few datapoints or a lack of variation within the dataset. The 'testDataset' function within RRP was used with a 98% reference range, produced from the wildtypes for each sex. ...

5) *Please explain more clearly the bases of your manual curation p.37 line 38 is stated "we eliminated genes where a single mid-range frequency showed a significant difference...". Isn't the lack of reproducibility of the results between animals carrying the same mutation an essential criterion of elimination? This is only indicated in the supplementary data and should be in the core of the manuscript. Identification of potential differences depending on the sex of the animal is also a very interesting result which should be found in the core of the manuscript rather than just in the supplementary data. Yet can a conclusion be reached with n=2? This is doubtful.*

In this revision we have moved the methods from Supplementary into the main body of the manuscript. Indeed, we agree with the reviewer that the manual curation criteria are an important aspect of the paper and so this section has been moved into the Results. Regarding the exclusion criterion-

"we eliminated genes where a single mid-range frequency showed a significant difference..."

this was not meant to refer to a lack of reproducibility of results between animals carrying the same mutation (as these would not have reached significance and therefore would not be shortlisted in the 328 genes), but instead refers to lines where all the mutant mice show an elevated single mid-range frequency. We have modified the text to make this more clear (p.7, line 10):

"In addition, strains that exhibited elevated thresholds at only one middle-range frequency (12, 18 or 24 kHz) were also removed from the list."

Finally, we agree with the reviewer that the identification of potential differences in hearing ability dependent on the sex of the mutant animal is an interesting result, and as suggested we have moved this to the core text of the revised manuscript. The two mutant lines concerned are *Eps8l1* and *Wdtd1*, for *Eps8l1* 2 males and 2 females were tested, and for *Wdtd1* 3 males and 4 females were tested. In both cases the elevated ABR thresholds of the females are very consistent within each line and the P values are highly significant (Supplementary table S1).

6) *Please rewrite abstract so it includes an actual summary of the results and methods used. Considering the limited body of proof presented, please refrain from using "deafness genes" along the manuscript and replace by "candidates genes".*

We have re-written the abstract to include a summary of the results and methods used, and throughout the manuscript we have indicated these are 'candidate' deafness genes.

Other comments

1) *Please consider changing "deafness" along the manuscript for "hearing loss" when more appropriate.*

We prefer to use the term *deafness*, as the term *hearing loss* could refer to mutants that display normal auditory function at the inception of hearing, but which is subsequently lost. Our phenotype assessment does not provide any information on whether or not there is true loss of hearing.

2) *Please change the first sentence of the abstract p. 4, line 1, it is not clear.*

We have now amended the first line of the abstract to read:

...The developmental and physiological complexity of the auditory system is likely reflected in the underlying set of genes involved in auditory function. ...

We believe that this is much clearer now.

3) *Please include in the introduction a brief description of how the null mutants were generated.*

We have added a comment in the second paragraph of the Introduction (p. 5), as well as a cross-reference to more detail in the Methods, that elaborates on the IMPC alleles that were generated and analysed.

4) *Measurement of ABR to a broadband click stimulus is mentioned in the introduction and the legend of several figures but the results are not presented, please either include these results or discuss why they were not included.*

Broadband clicks were analysed in some but not all centres, so we did not employ this data in our analysis. We have now removed all reference to the acquisition or analysis of the click data from the paper.

5) *ABR waveform analysis: was there any abnormalities in the waveform shape recorded for the different mutant mice? Any wave I latency change? Any marked and reproducible amplitude change?*

We have not yet carried out any detailed ABR waveform analysis. This was not a principal goal of this study in terms of identifying new candidate deafness loci. However, we recognise for the future that it may be valuable to undertake a comprehensive analysis of ABR waveforms for all 3006 mutants generated and analysed. Such an analysis might uncover further candidate deafness loci.

6) *Please define "QC" when first used p. 4, line 19*

We have defined the initials QC at first use (p.6, line 8).

7) *Here the results identify 52 new genes and 15 known genes. It is not clear to me that emphasizing that only 9 out of 261 genes excluded are known deafness genes is a good criterion: by comparison with the results presented here we could "miss" a large number of new genes.*

As the paper has been revised, we now have the opportunity to expand on our analysis of the false negative rate (see p.7 and p.8). Of the 261 excluded genes, only 9 had evidence for an auditory phenotype in mouse or human, but, importantly, for 6 of these there was only evidence in human and studies of mice had failed to uncover an auditory phenotype. Thus the false negative rate for known loci is 3/328 or around 1%. We now conclude (p.8, line 18) that extrapolating to novel loci:

...we can surmise that just over 3% or around 10 loci would be undiscovered from the set of 328 genes with putative auditory phenotypes identified from our analysis of 3006 genes.

8) P. 5 line 20 please correct to "The products of the 52 novel deafness genes..."

This has been corrected (p.9, line 15).

9) P. 5 line 11: precise which criteria were used to classify the genes in these 4 classes of hearing loss in the core of the manuscript.

In the revised manuscript we have had the opportunity to expand upon the criteria used to classify the genes to 'deafness' classes. On page 8, within the section titled 'Candidate deafness gene classification' we have included the following text (p.8, line 2):

...we assigned each of the 67 candidate genes to four broad classes of deafness using the following criteria: only the higher frequencies affected – High-frequency hearing loss (13 genes); only the lower frequencies affected – Low-frequency hearing loss (10 genes); if 3 or more frequency thresholds are ≥ 45 dB above the centre median – Severe hearing loss (25 genes); and, if less than 3 frequency thresholds are ≥ 45 dB above the centre median – Mild hearing loss (19 genes)...

10) P. 6 lines 10 to 25 This discussion should include the estimation of the number of genes implicated in syndromic deafness (which is at least 400). Those may be a better reference for comparison with the results obtained here.

The reviewer makes an important point, which we very much welcome and we have added significantly to the discussion to address this issue, and have included figures for both non-syndromic and syndromic loci (emphasising for the latter the paucity of knowledge on underlying genes). We now conclude (p.12, line 6):

In our screen for auditory-related genes we have explored around 15% of the mouse genome. Thus, we calculate that at a minimum the mammalian genome carries around 450 genes required for hearing function. In humans, around 150 non-syndromic deafness (DFNA, DFNB and DFNX) loci have been mapped, with 103 underlying genes identified (www.hereditaryhearingloss.org). There are over 400 genetic syndromes that include a deafness component^{1, 2}. However, most of the genes underlying syndromic deafness are unknown⁵. Our analysis thus reveals a very extensive and unexplored genetic landscape of genes required for auditory function and uncovers a large set of novel candidate deafness genes. We should emphasise that 450 genes represents a minimum set as our screen may have failed to uncover several classes of genes which contribute to hearing in either mouse or human...

We believe, reflecting the reviewer's comment, that this better reflects the comparative analysis.

11) For a long time, the presence of vestibular defects easily visualized (circling behavior...) prompted the test for potential hearing deficit in mutant mice. Do you have any data regarding possible vestibular defects in the mutant mice presented here? Please include data if available or comment.

As indicated above, the CSD test includes a visual inspection of vestibular phenotypes. We have interrogated the CSD data available for the 67 candidate deafness models and only 3 lines have a significant phenodeviance ($P < 0.001$) for the parameter Head bobbing/Circling. They are 3 known deafness genes, *Elmod1*, *Myo7a* and *Ush1c*. We have indicated this information in Table 2 and have added the following text to the manuscript (p.8, line 38):

...Auditory mutants can sometimes also display a vestibular component to their abnormal phenotype. The CSD test includes a visual inspection for behavioural signs that may indicate vestibular dysfunction e.g. head bobbing or circling. Only 3 of the 67 candidate deafness mutants were identified as having a head bobbing/circling phenotype. They are Elmod1, Myo7a and Ush1c, which are known hearing loss genes, with the respective allelic mutants roundabout, shaker-1 and deaf circler having previously been reported to exhibit vestibular dysfunction'. ...

12) *Some of the "new" genes identified, for example Atp2b1 (other name PMCA1) or Sema3f, are already known to have key function in the inner ear. Please discuss and reference appropriately.*

We have now made it clear in the Discussion, referring specifically to these two examples, that for some of the genes there is extant knowledge on expression in the inner ear or, for example, some aspect of developmental or physiological function in the cochlea. However, as we emphasise elsewhere in the paper, for all of the novel candidate deafness genes either for mouse or human, there is no specific data linking the gene to deafness. We have now stated in the Discussion (p.11, line 9):

Each of the 52 novel candidate deafness genes will merit further investigation. Some of the novel candidate genes are known to be expressed in the inner ear (e.g. Atp2b1) or to play a role in some aspect of hair cell function in the cochlea (e.g. Sema3f and sensory hair cell innervation)^{21, 22}. However, for the majority of novel candidate loci it will be important to further explore expression and function in the auditory apparatus.

Supplementary Data

1) *p.11 line 22: define SOPs*

We have defined the initials SOP at first use (p.13, line 35).

2) *p.12 line 25: use "article" instead of paper*

This has been changed (p.15, line 7).

3) *p.12 line 30: please correct or explain what you mean by "zygosity (either homozygous or heterozygous)" in this context.*

We have removed the text '*zygosity (either homozygous or heterozygous)*', which was included in error (p.15, line 13).

4) *Please indicate how the mice were anesthetized before ABRs.*

Each phenotyping centre may use a slightly different regime for anesthesia (agent and/or dose). This information is recorded in the metadata parameters and is available to view for each mutant on the portal. On page 14 we have included the following text (p.14, line 7):

'Importantly, each centre records significant metadata parameters including: equipment manufacturer; equipment model; recording environment; anesthetic agent; and, anesthetic dose.'

5) *Supplementary Table 4. Please add for each comparison the number of animals or "ears" included in the "Comparison".*

In the revised manuscript Supplementary Table 4 has become Supplementary Table 1, which has been edited to include the number of mutant mice for each model included in the comparison to the baseline wildtype control data.

Reviewer #2

This manuscript is an extremely valuable contribution. While the data are also freely searchable on the IMPC webpage, there was an extraordinary amount of work that went into the data generation.

We would like to thank the reviewer for their very supportive comments. We have addressed the very few comments.

Minor comments

1) *Page 5, line 4. There are two publications on neuroplastin in the inner ear. Please cite both.*

This has been done (p.9, line 33).

2) *Page 7, line 7. Remove "manuscript to be submitted."*

This has been removed.

3) *Fig. 1. The labels for the different centers should use the same font size as the other labels in the figure. They are very difficult to read. Do the authors wish to comment on the variability? Is this variability due to different instrumentation at each center?*

The labels have been edited for consistency. As the reviewer suggests, the variability evident likely reflects the different instrumentation and their environs at each centre.

4) *Suppl. Fig. 2. Looks pretty but does it really say much? The problem with inner ear genes is that many of them are not widely expressed and have not been studied before. There simply is a lack of information.*

Suppl. Fig. 2 is now Fig. 3 within the main body of the Results. It is important for the set of novel candidate deafness genes that we have identified to assess the depth of information available not just in terms of their expression within the inner ear but to what extent do we have knowledge about their interactions and networks. We believe that Fig. 3 is an important analysis and statement that confirms that as we continue to elaborate the genetic landscape and the set of genes involved with auditory function, there is indeed a very considerable dearth of knowledge. We have thus retained this Figure within the paper.

Reviewer #1 (Remarks to the Author):

We thank the authors for their responses and changes in the manuscript.

A few points still need attention:

- 1- You may want to reassess your estimation of false negative rate taking into account that from your own estimate in Introduction “To date, around two-thirds of the genes for non-syndromic deafness loci are known and, while the causative genes for several of the well-known deafness syndromes have been elucidated, the vast majority of genes underlying syndromes with deafness are unknown.” If as of now, only 3 out of the 328 genes considered, are known to be associated with hearing impairment in mice this does not mean that more are not involved. From your results only 15 out of the 67 genes you identified were known deafness-associated genes.
- 2- Fig. 3 does not add information to what is said in the text. If the authors want to present this figure they should do a manual curation of the information presented which will show them that a number of the pseudo protein interactions presented have not be validated experimentally.
- 3- The medical definition of “Hearing loss” is “any degree of impairment of the ability to apprehend sound.” This term does not imply the idea of change of hearing over time. It should be preferred to “deafness”.

Minor comments:

- 1- Consider changing in the Introduction line 146 “elucidated” by “identified” and line 154 “elaborate” by “identify”.
- 2- Although the authors stated “Broadband clicks were analysed in some but not all centres, so we did not employ this data in our analysis. We have now removed all reference to the acquisition or analysis of the click data from the paper.” Legends of Figures 1 and 3 still indicate lines 695-696: “The available broadband click ABR data for each gene is not provided in these plots.” and lines 704-705 “The available broadband click ABR data for each gene is not presented in these plots.” Please remove those sentences or present the data.
- 3- Line 270, you may want to add the possibility of a deficit at the level of the central auditory pathway.

Reviewer #2 (Remarks to the Author):

The authors addressed my concerns and those of reviewer #1 very thoroughly.

Reviewer #1

We thank the authors for their responses and changes in the manuscript.

We thank the reviewer for their supportive comments.

A few points still need attention:

1) *You may want to reassess your estimation of false negative rate taking into account that from your own estimate in Introduction "To date, around two-thirds of the genes for non-syndromic deafness loci are known and, while the causative genes for several of the well-known deafness syndromes have been elucidated, the vast majority of genes underlying syndromes with deafness are unknown." If as of now, only 3 out of the 328 genes considered, are known to be associated with hearing impairment in mice this does not mean that more are not involved. From your results only 15 out of the 67 genes you identified were known deafness-associated genes.*

We agree with the reviewer that due to the large number of unknown hearing loss genes we must account for the possibility of unknown deafness genes in estimating the false negative rate. We had indeed accounted for that in our calculation, but we realize this was not very clear in our explanation, and we omitted to sum the expected false negative rate for known and novel loci. As such, we took the original paragraph that read:

'Our investigation of the false negative rate (see above) allows us to make some assumptions about the number of known and novel loci that may have been missed by our stringent manual curation of the set of 328 genes for which there was initial evidence of an auditory phenotype. Overall, we found that for known loci 3/328 genes were false negatives or around 1%. Extrapolating to novel loci, we can surmise that just over 3% or around 10 loci would be undiscovered from the set of 328 genes with putative auditory phenotypes identified from our analysis of 3006 genes.'

And have edited it so that it now reads as:

'Our investigation of the false negative rate (see above) allows us to make some assumptions about the number of known and novel loci that may have been missed by our stringent manual curation of the set of 328 genes for which there was initial evidence of an auditory phenotype. Overall, we found that for known loci 3/328 genes were false negatives or around 1%. Extrapolating to novel loci, based on the ratio of novel to known loci (51/15) within the set of 67 candidate hearing loss genes, we surmise that around 10 (i.e. $3 \times 51/15$) additional loci would be undiscovered from the set of 328 genes with putative auditory phenotypes. This provides a minimum estimate of 13 undiscovered hearing loss loci from our analysis of 3006 genes.'

2) *Fig. 3 does not add information to what is said in the text. If the authors want to present this figure they should do a manual curation of the information presented which will show them that a number of the pseudo protein interactions presented have not be validated experimentally.*

We concur with the reviewer, and have now further refined the presentation to indicate the evidence base for each of the presented interactions. As such we have changed Figure 3 to include coloured edges to illustrate the type of interaction, and to demonstrate more clearly those more robust interactions based on experiment or curated databases (bold red edges, Fig. 3), as opposed to those that are based on evidence types that include text mining (thin grey edges, Fig. 3). We have altered the main text and figure legend accordingly:

Page 9, line 20

'We assessed for potential interactions between the encoded proteins of the 52 novel genes with known human hereditary hearing loss dominant (DFNA), recessive (DFNB) and X-linked (DFNX) gene products. The predicted network interaction map shows that a significant number of known genes (65) formed a densely connected hub that incorporated 11 IMPC novel genes (Fig. 3).'

Page 16, line 17

'Assessment of protein interactions was undertaken using STRING (version 10.0) at a combined medium confidence setting of ≥ 0.4 .'

Page 21, line 14

'Figure 3. A network interaction map incorporating the proteins encoded by the 67 hearing loss genes identified from the IMPC ABR test. A STRING interaction map, incorporating known and predicted interactions, was generated for the known (15) and novel (52) IMPC hearing loss genes. Blue nodes are the IMPC novel genes. Other nodes are previously reported genes that underlie hereditary hearing loss in humans, including DFNA (dominant hearing loss genes), DFNA/DFNAB (hearing loss genes showing both dominant and recessive inheritance), DFNB (recessive hearing loss genes), and DFNX (X-linked hearing loss genes). IMPC known genes are a subset of these genes and are highlighted by purple colour. The highly connected interaction map (shaded) consists of 65 known hearing loss genes and 11 novel IMPC genes. The majority of IMPC novel genes (41) are unconnected to the central network. Thin grey edges show interactions with a combined confidence score of ≥ 0.4 , summated from evidence types: 'curated databases'; 'experimentally determined'; and, 'automated text mining'. Bold red edges show 'known' interactions with a combined confidence score of ≥ 0.4 , summated from 'curated databases' and 'experimentally determined' only.'

3) The medical definition of "Hearing loss" is "any degree of impairment of the ability to apprehend sound." This term does not imply the idea of change of hearing over time. It should be preferred to "deafness".

As requested, we have substituted the term 'deafness' with the words 'hearing loss' throughout the manuscript, including figures, tables and legends.

Minor comments

1) Consider changing in the Introduction line 146 "elucidated" by "identified" and line 154 "elaborate" by "identify".

These have been changed.

2) Although the authors stated "Broadband clicks were analysed in some but not all centres, so we did not employ this data in our analysis. We have now removed all reference to the acquisition or analysis of the click data from the paper." Legends of Figures 1 and 3 still indicate lines 695-696: "The available broadband click ABR data for each gene is not provided in these plots." and lines 704-705 "The available broadband click ABR data for each gene is not presented in these plots." Please remove those sentences or present the data.

As requested, reference to click ABR data has been removed from the legends of Figures 1 and 2.

3) *Line 270, you may want to add the possibility of a deficit at the level of the central auditory pathway.*

We have added this comment. The sentence now reads:

'Without further investigation of individual mutants, we are unable to distinguish between hearing loss due to a deficit at the level of the central auditory pathway, a sensorineural origin in the inner ear, or due to conductive hearing loss due to middle ear problems, such as otitis media.'

Reviewer #2

The authors addressed my concerns and those of reviewer #1 very thoroughly.

We thank the reviewer for their very supportive comments.

Reviewer #1 (Remarks to the Author):

The authors report a very important work which will significantly impact the field. The authors addressed my concerns thoroughly, yet there is still a mistake in the false negative rate which should be corrected. I was hoping the authors would realize it in their corrections after my last review but I was not explicit enough in my comments.

The false negative rate of the authors' manual curation done on 328 genes, is not $3/328$. The false negative rate is the percentage of mice with hearing loss who incorrectly receive a negative test result. Therefore, the false negative rate of the manual curation should be at least $100 \cdot 3 / (67 + 3)$ assuming that all the 67 mouse lines identified as positive for hearing loss really have hearing loss and are not a mix of true and false positive.

The false omission rate of the manual curation may be what the authors are trying to point at here, in this case it is at least $3/261$.

If the authors wish to extrapolate using their results 52 should be used instead of 51 in their calculations ($67 - 15$).

Although the performance of the overall screening test of 3006 mouse lines is probably the most relevant, it is impossible to compute any statistics such as false negative rate or false omission rate with respect to the overall procedure only studying the number of known hearing loss genes among the 328 genes which were first identified as potentially linked to hearing loss. If the authors want to mention a false omission rate on their overall test on 3006 mouse lines they should look at how many of these genes are known hearing loss genes among all the $3006 - 67$ genes classified as negative in their tests, and the statistics should be re-computed accordingly. I am not asking for that as there is too many unknown.

Reviewers' comments

Reviewer 1:

The authors report a very important work which will significantly impact the field. The authors addressed my concerns thoroughly, yet there is still a mistake in the false negative rate which should be corrected. I was hoping the authors would realize it in their corrections after my last review but I was not explicit enough in my comments.

The false negative rate of the authors' manual curation done on 328 genes, is not 3/328. The false negative rate is the percentage of mice with hearing loss who incorrectly receive a negative test result. Therefore, the false negative rate of the manual curation should be at least $100 \times 3 / (67 + 3)$ assuming that all the 67 mouse lines identified as positive for hearing loss really have hearing loss and are not a mix of true and false positive. The false omission rate of the manual curation may be what the authors are trying to point at here, in this case it is at least 3/261. If the authors wish to extrapolate using their results 52 should be used instead of 51 in their calculations (67-15).

Although the performance of the overall screening test of 3006 mouse lines is probably the most relevant, it is impossible to compute any statistics such as false negative rate or false omission rate with respect to the overall procedure only studying the number of known hearing loss genes among the 328 genes which were first identified as potentially linked to hearing loss. If the authors want to mention a false omission rate on their overall test on 3006 mouse lines they should look at how many of these genes are known hearing loss genes among all the 3006 - 67 genes classified as negative in their tests, and the statistics should be re-computed accordingly. I am not asking for that as there is too many unknown.

Firstly, we thank Reviewer 1 for their very positive response to our revised manuscript, and welcome the very helpful clarification of their earlier thoughts from the previous review.

It is indeed only an issue of reporting "false omission rate" as opposed to "false negative rate", and there is no technical disagreement. We agree with the reviewer that we should have referred to our calculation as the false omission rate. Moreover, we agree with the reviewer that we should have used 261 as the denominator in the calculation, and this estimate of FOR should not apply across all 3006 genes. The use of 51 in our text was simply a typographical error – it should of course be 52 as the reviewer states. We are grateful to you and the reviewer for the very helpful comments in terms of clarifying this point.

Previously the paragraph read as:

'Our investigation of the false negative rate (see above) allows us to make some assumptions about the number of known and novel loci that may have been missed by our stringent manual curation of the set of 328 genes for which there was initial evidence of an auditory phenotype. Overall, we found that for known loci 3/328 genes were false negatives or around 1%. Extrapolating to novel loci, based on the ratio of novel to known loci (51/15) within the set of 67 candidate hearing loss genes, we surmise that around 10 (i.e $3 \times 51/15$) additional loci would be undiscovered from the set of 328 genes with putative auditory phenotypes. This provides a minimum estimate of 13 undiscovered hearing loss loci from our analysis of 3006 genes.'

This has now been corrected such that the paragraph now reads:

'Our investigation of false negatives (see above) allows us to calculate a false omission rate (FOR) for loci that may have been missed by our stringent manual curation of the set of 261 genes for which there was initial evidence of an auditory phenotype. Overall, the FOR for known loci is 3/261 or around 1%. Extrapolating to novel loci, based on the ratio of novel to known loci (52/15) within the set of 67 candidate hearing loss genes, we surmise that around 10 (i.e. $3 \times 52/15$) additional loci would be undiscovered from the set of 261 genes. This provides an estimate of 13/261 for the FOR.'

In addition, we have edited the Discussion. Specifically, the sentence that read:

'First, we discuss above our estimate of the false negative rate arising from our stringent manual curation of the set of 328 candidate genes.'

Has been changed so that it now reads:

'First, we discuss above our estimate of the false omission rate arising from our stringent manual curation of the set of 328 candidate genes.'